# Waste Aluminum Application as Energy Valorization for Hydrogen Fuel Cells for Mobile Low Power Machines Applications

**DOI:** 10.3390/ma14237323

**Published:** 2021-11-30

**Authors:** Xavier Salueña-Berna, Marc Marín-Genescà, Lluís Massagués Vidal, José M. Dagà-Monmany

**Affiliations:** 1Mechanical Engineering Department, ESEIAAT-UPC, Colom 1, 08222 Terrassa, Spain; xavier.saluena@upc.edu; 2Mechanical Engineering Department, ETSEQ-URV, Països Catalans 26, 43007 Tarragona, Spain; 3Electrical Engineering Department, ETSE-URV, Països Catalans 26, 43007 Tarragona, Spain; lluis.massagues@urv.cat; 4Chemical Engineering Department, ESEIAAT-UPC, Colom 1, 08222 Terrassa, Spain; jose.maria.daga@upc.edu

**Keywords:** hydrogen, aluminum waste, isopropyl alcohol, hydrogen generation, hydrogen flow rate, sustainability, fuel cell, hydrogen machines

## Abstract

This article proposes a new model of power supply for mobile low power machines applications, between 10 W and 30 W, such as radio-controlled (RC) electric cars. This power supply is based on general hydrogen from residual aluminum and water with NaOH, so it is proposed energy valorization of aluminum waste. In the present research, a theoretical model allows us to predict the requested aluminum surface and the required flow of hydrogen has been developed, also considering, in addition to the geometry and purity of the material, two key variables as the temperature and the molarity of the alkaline solution used in the hydrogen production process. Focusing on hydrogen production, isopropyl alcohol plays a key role in the reactor’s fuel cell vehicle as it filters out NaOH particles and maintains a constant flow of hydrogen for the operation of the machine, keeping the reactor temperature controlled. Finally, a comparison of the theoretical and experimental data has been used to validate the developed model using aluminum sheets from ring cans to generate hydrogen, which will be used as a source of hydrogen in a power fuel cell of an RC car. Finally, the manuscript shows the parts of the vehicle’s powertrain, its behavior, and mode of operation.

## 1. Introduction

There are low-power machines such as radio-controlled electric cars on the market with a fuel cell. The advantage over battery machines is the shorter electric recharging times, which makes their autonomy longer [1]. The hydrogen storage necessary for low-power machines’ fuel cells is usually carried out with metal hydride tanks or with small metal pressurized hydrogen tanks [2], and the generation of hydrogen is by electrolysis, in an external device from electricity from the electrical network [3], from biological production [4], by fossil fuel reforming [5,6] or pyrolysis from gas natural, heavy oils, naphtha or coal, and the hydrogen is distributed commercially from pressure cylinders [7,8]. Researchers have proposed other methods to obtain hydrogen by hydrolysis of metal (aluminum or magnesium) or metal hydride [9]. There are currently a lot of studies in chemical hydrides, including ammonia [10], ammonia borane, metal boron hydride, formic acid, hydrazine hydrate, aromatic compounds, or sodium borohydride [11] to obtain it in situ [12]. There is even an article where ammonia borane is used to feed a fuel cell of 30 W, but not in a machine due to the complexity of this method to be installed on small machines [13].

One of the current methods to facilitate the aluminum–water reaction from liquid aluminum particles uses gallium [14] or activated Ga-In-Sn-Bi alloys [15], but these systems require previous preparations, are expensive and the associated wastes must be processed [16]. The other methods used to facilitate the aluminum–water reaction consist of the aluminum introduction into an alkaline (or acid) solution to eliminate the surface oxide layer and subsequently the posterior layers of aluminum hydroxide that are formed. In some cases, promoters, such as oxides and salts, are used to reduce the amount of alkali needed, but in some cases the chemical reactor should be warmed up [17] at the startup, and sometimes the precipitated residue must be processed in recycling [18]. 

In this article, a sustainable method is proposed to generate high purity hydrogen, in situ, from cheap waste aluminum sheets, such as rings cans, and an alkaline solution by the aluminum–water reaction. The residue (sodium aluminate) from this system is not only fully recyclable, but with renewable energy, it can be transformed into the original products (aluminum, NaOH and water) since no other catalyst is used [19]. 

In the bibliography, there are several cases of industrial prototypes where this technology is applied, although it usually starts with high purity aluminum powder in the size order of microns and is very expensive [20]. In such prototypes, aluminum is usually dosed, which enters the reactor, to control and predict the hydrogen flow produced [21]. Generally, in fuel cells, it is required that said flow to be constant over time [22] since they work at nominal power in parallel with an auxiliary battery, which absorbs the electrical energy excess generated, not consumed, and besides, it regulates the voltage [23].

Technologically, and more in a small vehicle, it is difficult to dose a solid to control the flow [24], so initially, all the aluminum and the alkaline solution are inside the reactor. When aluminum reacts with water in an alkaline medium, the reaction is exothermic [25] and the temperature rises, so a no constant flow rate of hydrogen is produced because the flow rate increases strongly with the temperature [26]. When operating in this way, it is necessary to have an intermediate tank to store the surplus hydrogen, which is the difference between the flow rate generated by the reactor and that consumed by the machine fuel cell. Another drawback of the aluminum–water reaction in an alkaline medium is that although the hydrogen obtained is of high purity [27], it tends to carry NaOH particles and water vapor [28], which could alter the operation and durability of the fuel cell [29].

The current article proposes an alternative method for obtaining hydrogen in an alcoholic medium (isopropyl alcohol), in a controlled manner. In another article, we explain how we tested different alcohols to check which was the optimal one [30]. A model obtained in a theoretical-experimental way is proposed, capable of characterizing the hydrogen flow according to the: chip dimensions, the aluminum active surface, solution molarity and temperature. Likewise, it describes how alcohol affects the reaction, carrying out a comparative analysis of the obtained hydrogen flow rates and analyzing the purity of the hydrogen. Finally, the knowledge acquired in the accurate feeding with soda rings a fuel cell of a radio-controlled car, called dAlH2orean, has been applied.

The main novelty of this work is the simplification of the hydrogen generator, thanks to alcohol, which allows it to be mounted in a small low-power machine since no heavy exchangers are needed to keep the flow controlled and neither are expensive filters to eliminate the contaminants that can damage the fuel cell and its application. 

## 2. Materials, Theoretical Models and Tests

### 2.1. Materials

The aluminum used to obtain the theoretical-experimental model consists of a 0.5 mm thick aluminum sheet according to ISO209: 2007 for aluminum with 99.5% purity. The aluminum is the same as that used in the manufacturing of soda rings. A soda ring is nothing more than an aluminum sheet of constant thickness, cut and folded, so the behavior will be similar. Regarding sodium hydroxide, the analysis was developed with a solution of sodium hydroxide from 1 M to 10 M in test 1 and 7.5 M in the rest of the tests, starting from 50% *w*/*w* solutions (PANREAC 142404.0716) and distilled water.

Finally, regarding alcohol, isopropyl with a 99% purity has been used. The boiling alcohol temperature must be higher than 70 °C, and since the chemical reaction is exothermic if alcohol is not used, the reactor temperature could surpass the 80 °C, and if the alcohol boiling point was inferior to that of isopropyl alcohol, it could completely vaporize at a lower temperature and not perform its function. Because of the isopropyl alcohol presence, the temperature remains in the range from 50 °C to 60 °C inside the reactor. At that temperature range, bayerite is formed [31], and if it were higher, boehmite would form and the theoretical model used could be not appropriated.

### 2.2. Theoretical Kinetic Model

Chemical Reactions.

The chemical reaction to obtain hydrogen from water–aluminum mix [32] is obtained by next Equation (1).
2Al (s) + 6 H_2_O (L) → 2 Al (OH)_3_ (s) + 3H_2_ (g) (1)

Solid Al (OH)_3_ creates a surface layer that prevents water from reacting with the aluminum, becoming passivated solution. In order to avoid passivation [33], the reaction will be performed by an NaOH alkaline solution, and finally Al (OH)_3_ is transformed into aqueous Na Al (OH)_4_, leaving Equation (2).
2Al (s) + 6 H_2_O (L) + 2 NaOH (aq) → 2 Na Al (OH)_4_ (aq) + 3H_2_ (g) (2)

According to the bibliography [34], the hydrogen flow rate, and therefore the reduction in the mass of the sheet, is proportional to the apparent rate constant of the reaction k. The hydrogen flow rate is also proportional to the active surface of the aluminum and the concentration of the solution. To formulate the kinetic model, it is assumed that aluminum is the limiting reactant, and therefore the flow rate will depend on the active surface of the aluminum. This hypothesis serves only as a starting point for the study since the surface can be reduced by its partial passivation. The active surface decreases according to time since aluminum is transformed into Al (OH)_3,_ and also depending on the shape of the original chip or specimen. One of the contributions of the present research is the modeling of the active surface for the form of a simple sheet prismatic shape, and which the size is needed to produce the minimum hydrogen flow rate to be consumed by the fuel cell.

#### 2.2.1. Theoretical Hydrogen Flow Rate 

The previewed hydrogen flow rate *Q*_*H*_2_,*t*_ is obtained across Equation (1) as a function of the aluminum active surface *(S_Al,t_)*, the thickness reduction per surface as time function *(e)*, the weight *M_Al_* and density *ρ_Al_*, assuming ideal gas behavior, can be expressed according to Equation (3), [35].
(3)$$Q_{H_{2},t}=\frac{3}{2}\frac{\rho_{Al}}{M_{Al}}\left(\frac{RT}{P}\right)(S_{Al,t}\cdot e)$$

#### 2.2.2. Aluminum Active Surface

The material is aluminum waste in sheet form. A model has been developed, which describes the active surface of the aluminum foil as a function of time, assuming that the foil is ideally prismatic. An aluminum sheet (Figure 1) of measure *L·a·p* is considered, where the thickness: *p*, *a* is the width and length is *L.* The speed at which aluminum consumes is called corrosion rate *Rc.*

We assume that there is a reduction in the thickness per surface, *e* (mm/minutes), that depends on time *t*. This is applied to all the sheet surfaces (Figure 1) until the thickness *p* is used up, which will be less.

The active surface of the sheet as a function of the initial aluminum mass *W*_*Al*,0_, purity *η* and density *ρ* is described by Equation (4).
(4)$$S_{Al,t}=\frac{\;2W_{Al,0}\eta }{\rho_{Al}\;L\;a\;p}\left[\left(L^{\prime }+a^{\prime }+2e\right)*\left(p^{\prime }+2e\right)+\left(L^{\prime }*a^{\prime }\right)\right]$$

Logically, the sheet lifetime, from when the beginning to react until it is totally consumed, depends on its smallest size *p*, so, its thickness. As the thickness of the sheet is very small, the reduction suffered by the upper and lower faces, *L*, *a*, is minimal, so that the active surface would suddenly go from a considerable value to zero when the thickness *p′* = *p* − 2*et* equals zero. This happens gradually since in the last phase the sheet floats and disappears from the edges towards the center. The obtained flow rate as time function *Q*_*H*_2_,*t*_, for a sheet, completing Equation (3) with Equation (4), is determined by Equation (5).
(5)$$Q_{H_{2},t}=\frac{3W_{Al,0}\eta }{M_{Al}\;L\;a\;p}\left(\frac{RT}{P}\right)-\left[\left(L^{\prime }+a^{\prime }+2e\right)*\left(p^{\prime }+2e\right)+\left(L^{\prime }*a^{\prime }\right)\right]\;\cdot e$$

Being the theoretical volume of hydrogen *V*_*H*_2__, produced in a time *t* = *n* [minutes], according to Equation (6):(6)$$V_{H_{2}}=\frac{3}{2}\frac{\rho_{Al}}{M_{Al}}\left(\frac{RT}{P}\right)\sum_{t=0}^{n}(S_{Al,t}\cdot e)$$

The volume of hydrogen (theoretical) *V*_*H*_2__ for one sheet is given by Equation (7):(7)$$V_{H_{2}}=\frac{3W_{Al,0}\eta }{M_{Al}\;L\;a\;p}\left(\frac{RT}{P}\right)-\sum_{t=0}^{n}\left[\left(L^{\prime }+a^{\prime }+2e\right)*\left(p^{\prime }+2e\right)+\left(L^{\prime }*a^{\prime }\right)\right]\;\cdot e$$

The calculation of the thickness reduction per surface *e* will be made from the determination of the corrosion rate *Rc*, which depends on the volume of hydrogen (mL) given in Equation (7), the surface (cm^2^) Equation (4) and time *t* (min).
(8)$$R_{c}=\frac{V_{H_{2}}}{S\cdot t}$$

*Rc* depends on the molarity of the solution and the reaction temperature. For each molarity, its value can be determined as a function of temperature by using the Arrhenius equation, Equation (9), where *A* is the frequency coefficient (mL cm^−2^ min^−1^), *E_a_* is the activation energy (kJ mol^−1^), *R* is the ideal gas constant (kJ K^−1^ mol^−1^) and *T* is the temperature (K).
(9)$$lnRc=lnA-\frac{Ea}{RT}$$

The value of *Rc* has been found experimentally from the volume of hydrogen generated. The equation has been found experimentally, through the characteristic graph of the Arrhenius linear model, lnk (min^−1^) vs. 1/*T* (K^−1^), and comparing the values obtained experimentally with those given in the bibliography [36,37,38], and the accepted values are in the range between 42.5 and 68.4 kJ/mol. From the mean *Rc*, through Equation (8), the theoretical thickness reduction *e* can be calculated for any temperature and molarity inside the studied ranges.

### 2.3. Validation Tests 

Validation of the theoretical model has been performed by using several tests that have been developed to validate the model, that is, to understand how Al 99.5 aluminum sheets react in a caustic soda solution at different molarities, with and without isopropyl alcohol. The tests were carried out in a 500 mL flask at a constant temperature of 25 °C, 40 °C and 60 °C; 200 mL in a solution of NaOH (with or without 100 mL of isopropyl alcohol, depending on the test) were introduced into the flask. The aluminum specimens were then introduced into the flask keeping a constant temperature, and the flask was hermetically capped, connected to a flowmeter. The aluminum reacted with the solution generating hydrogen, which was cooled to a temperature of 15 °C before the flow measurement. The test ended when all the aluminum was consumed. Two tests were developed with each sample, obtaining the mean values, and repeating the measurement if the variation exceeded 90%. In some of the tests, the dimensions the thickness of the sheets was measured at one-minute intervals with a precision micrometer. We have applied the one-factor-a-time strategy because the behavior of the thickness reduction as a function of time is not constant over time and the only aim is to find the optimal conditions to achieve the maximum flow rate. 

#### 2.3.1. Test 1: Effect of Molarity

Through test 1, it is intended to find experimentally how the reaction behaves as a function of molarity to corroborate Equation (5) and to find the optimal molarity that can decompose the aluminum plate in the shortest possible time to generate the highest hydrogen flow rate. 

Aluminum sheets of 20 × 30 × 0.5 mm^3^ were tested at 25 °C in NaOH solutions at different molarities (1 M, 2 M, 3 M, 4 M, 5 M, 6 M, 7 M, 8 M, 9 M, and 10 M). Since the best results have been given for the molarity of 7 and 8 M, the following tests were carried out at 7.5 M. 

#### 2.3.2. Test 2: Effect of the Dimension of the Sheet

Using test 2, two experiments were tested to compare the hydrogen generation in the NaOH solution (7.5 M) at 25 °C temperature in sheets (foil format) or chips (foil cuttings) to check if the sheets, as in the case of rings, would have the same behavior observed in chips. A 20 × 30 × 0.5 mm^3^ sheet was first tested and later a set of 104 chips of 2.002 × 2.002 × 0.5 mm^3^, so that in the two experiments, both specimens had the same initial theoretical contact surface (1250 mm^2^). Since the purity of the aluminum was 99.5%, the initial area was 1243.75 mm^2^.

#### 2.3.3. Test 3: Temperature Effect 

Through test 3, the temperature effect in the generation of the hydrogen flow was compared, as well as in the reduction of the thickness of the sheet, to be able to validate Equation (7). Given the same behavior of the sheets or chips, a 20 × 30 × 0.5 mm^3^ sheet in 7.5 M NaOH solution was used for the test, and we experimented how it would be affected for the temperatures of 25 °C, 40 °C and 60 °C. After that, we performed the experiment with a ring can for the temperatures of 50 °C and 60 °C in a 4 M NaOH solution.

#### 2.3.4. Test 4: Effect of the Addition of Alcohol

By test 4, the effect of the addition of alcohol isopropyl in 7.5 M NaOH solution was compared. For this, a 20 × 30 × 0.5 mm^3^ aluminum foil was processed at 60 °C temperature, with and then without isopropyl alcohol.

#### 2.3.5. Test 5: Hydrogen Purity Analysis

To obtain the hydrogen percentage, some specimens of one hydrogen liter were generated and then collected, after a purge carried out by a generation of the same gas, in a Tedlar bag. The hydrogen obtained was analyzed through a “Agilent 3000 A” micro gas chromatograph equipped with 3 independent modules, each provided with a thermal conductivity detector (TDC) and columns to measure light gases (molecular sieve), light hydrocarbons (with Plot U) and oxygenates (Stabil wax). The calibration of the apparatus was made from the hydrogen of 99.999% purity. Three analyses were performed for each of the collected samples. 

#### 2.3.6. Test 6: Application in a Vehicle Prototype with Hydrogen Fuel Cell 

To demonstrate the validity of the model and its application in the field of engineering, a radio-controlled vehicle has been developed (Figure 2), with a fuel cell whose hydrogen has been generated from aluminum soda rings.

The fuel cell H-12 used is of the proton exchange membrane (PEM)-type from the company Horizon, whose maximum flow required at 1.5 A–7.8 V [11.7 W] operating power is 150 mL/min with 99.995% purity hydrogen, according to manufacturer’s specifications. As the generation system is not completely hermetic and may contain some humidity, the hydrogen is not pure, and therefore it is necessary to leave the outlet of the fuel cell open so that air or humidity does not accumulate inside it because the performance of the battery may decrease. In the 225 mL PVC tank, 15 soda rings of 0.285 g each, 100 mL of 4 M NaOH solution and 30 mL of isopropyl alcohol, both preheated to 40 °C, were introduced. Hydrogen, which carried some alcohol vapor, was filtered with a diffuser filter containing 50 mL of distilled water. At the outlet of the water filter, the hydrogen was dried using silica-gel 3-6 211335.1210 Panreac and introduced into the fuel cell. The fuel cell was installed in parallel, with a protection diode, with a battery made up of 6 AA rechargeable NiMH 1.2 V and 1900 mAh batteries. Both powered a Bycmo Speed-PRO variable speed drive, the radio control receiver, a Mabuchi RS-540SH 4.8–7.2 V motor (Mabuchi Motor Company, Matsudo, Japan) and the servo motor for the steering. In the experiment, the hydrogen flow generated at the filter outlet was measured.

## 3. Results 

### 3.1. Trials

#### 3.1.1. Effect of Molarity

In Figure 3, the reduction of the thickness of the sheet per hour is described over 3 h, at which time the sheet reacts at different molarities solution and 25 °C. We have chosen approximately 3 h because in the fastest of those experiments studied, the 7 M sheet is about to disappear, due to its small thickness of 0.02 mm. 

#### 3.1.2. Effect of Chip Size

In the first test, the hydrogen generation from aluminum foil was analyzed, while in this second experiment 104 chips of the same initial active surface as the foil were analyzed, at 7.5 M, with alcohol and at 25 °C. The result is shown in Figure 4, in addition to the theoretical values obtained according to Equation (9).

In Figure 4, it is observed on the top graph how the hydrogen flow rate generated in the sheet is constant, staying between 5 and 4 mL/min, while with the chips the flow rate decreases more rapidly. This fact, as can be seen on the lower graph, is explained by the active surface, and given Equation (4), it is seen that it is reduced faster in the chips than in the sheet. Initially, for both typologies, there is enough active aluminum surface that the limiting factor is the water flow and thus the flow follows Equation (3). In the chips, after approximately half an hour (30 min), the active surface is less than critical and the hydrogen flow then follows Equation (7), where the limiting factor is aluminum. It is checked that the time reaction depends on the thickness (p variable) in both formats.

#### 3.1.3. Effect of Temperature

In Equation (9), the effect of the temperature [39] has been validated by introducing aluminum sheets measuring 20 × 30 × 0.5 mm^3^ in a NaOH solution (7.5 M), maintaining the constant temperature at 25 °C, 40 °C and 60 °C, observing that the theoretical active surface given in Equation (9) is perfectly adapted to the experimental one. In this case, the activation energy is *E_a_* = 56.99 kJ mol^−1^. The comparison between the theoretical and experimental reduction is shown in Table 1. 

In the case of ring cans in a 4 M NaOH solution, the temperature was kept constant at 50 °C and 60 °C. The comparison between the theoretical and experimental reduction is shown in Table 2.

#### 3.1.4. Adding Alcohol Effect

Figure 5 shows the effect of isopropyl alcohol addition to NaOH solution. It worked with an aluminum sheet of 20 × 30 × 0.5 mm^3^ at 60 °C.

The alcohol used has a lower density than the NaOH solution 7.5 M [40]. Initially there are two phases; the upper phase is alcohol and the lower is soda solution. The sheets are submerged in the NaOH phase. As the aluminum–water reaction is exothermic, it increases the reactor temperature, and the hydrogen flow rate increases due to the corrosion rate and thickness reduction increase exponentially, too, as shown in table (Table 1). To achieve a constant flow, the system under control must ensure that the temperature is also constant. As the alcohol evaporates it maintains the temperature close to the boiling point. The other advantage of using alcohol is that the reaction can be controlled and stopped, as it does not react with aluminum, by controlling the alcohol level so that the aluminum is submerged in it. Moreover, the alcohol works as a reactive filter and does not allow the hydrogen to carry away the NaOH particles that can contaminate the fuel cell. Usually, to return the vaporized alcohol to the reactor, a condenser must be used [41].

#### 3.1.5. Hydrogen Purity

The purity of the hydrogen produced with the sheets has been analyzed in anticipation that the aluminum residues could contain volatile or reactive elements that generate some undesirable gas. Gases such as CO, C1-C6 hydrocarbons or C1-C3 oxygenates were not detected on the micro gas chromatograph, nor were any alkali detected by pH analysis. For the flow generated, the purity of the hydrogen is greater than 98% in all the cases, with impurities of air (<0.06%), alcohol (<0.007%) and water (>1.28%). This is due to the lack of tightness of the system connections since the equipment is made of glass or plastic and silicone tubes, which allows the passage of air into the circuit and the use of a water filter that generates more humidity than the dryer could absorb. The obtained alcohol after filtering, for hydrogen flow rates below 3 L/min, is not relevant (<0.007%), but in the course of time, it is deposited in the filter and due to its low density, it can be dragged upstream inside the fuel cell.

#### 3.1.6. Prototype of a Vehicle with Hydrogen Fuel Cell from Aluminum Soda Rings

In Figure 6, the flow calculated in the theoretical model is compared with the experimental flow produced by the 15 rings with 4 M NaOH solution. In the model, the rings have been simplified as sheets of 16 × 25 mm^2^ with two holes with a radius of 5.1 mm and 4.7 mm thick. In the experiment, the temperature inside the generation tank and the intensity generated by the fuel cell at the working voltage was measured. It is verified that the flow obtained adapts to the theoretical model and how the alcohol regulates the start of the reaction, we have reduced to 4 M the molarity to adapt the system to the required flow rate. 

## 4. Discussion

About molarity, between 1 M to 7.5 M, the reduction increases with the molarity. It is observed that between 1 M and 4 M, the decrease in thickness is constant, but from 5 M, it increases non-linearly to 7.5 M. Between 7.5 M and 10 M, it is reduced. According to the bibliography, it is because an increase in molarity leads to an increase in conductivity, which facilitates the reaction, but also leads to an increase in viscosity that harms the hydrogen generation [42]. It follows that the optimal point is at 7.5 M.

On the other hand, the alcohol is not capable of completely controlling the temperature due to the exothermic aluminum–water reaction. Since there is no condensation system in the vehicle, this prevents, in this case, the temperature from exceeding 68 °C; however, without alcohol, the hydrogen flow increases uncontrollably as it quickly exceeds 80 °C and the cap of the tank is opened due to overpressure.

In the fuel cell, since the hydrogen is not completely pure, a power less than the expected 1.4 A is produced at 7.8 V [10.9 W] when the flow rate exceeds the 150 mL/min required according to the manufacturer. If all the hydrogen generated was used by the fuel cells, the performance waste aluminum to hydrogen is 98% and the performance hydrogen to fuel cell power is around 93%. The total balance energy performance is 91%. In this RC car, the fuel cell outlet must be left open so that excess hydrogen is lost but could be stored by pressure. It is sufficient to charge the battery with the engine stopped and to help it while the vehicle is running, increasing its autonomy. At maximum power, the vehicle’s engine consumes 2.4 A at 7.8 V, so thanks to the fuel cell, the auxiliary battery needs to supply only 1 A at maximum power. The duration of the rings is approximately 30 min in operation, being 16 min at maximum power.

## 5. Conclusions

In this article, a new method is described to obtain high purity hydrogen from aluminum waste, at a constant flow rate, regardless of the amount of aluminum product in the fuel cell reactor, using isopropyl alcohol. 

A theoretical model has been described that allows us to find the flow rate and the volume of hydrogen generated as a function of purity and dimensions of the aluminum sheets, as well as the molarity and temperature of the NaOH solution. To find the optimal solution, it is necessary to set the molarity and to determine the geometry and the corrosion rate of the material as the function of the temperature in each case. 

The behavior of the aluminum–water reaction in the presence of alcohol at 25 °C at different molarities has been determined and described and it has been determined that the optimal concentration for maximum flow rate is 7.5 M. For molarities, between 1 M and 4 M, the flow of hydrogen generated is constant because the reduction in thickness is constant in the course of time for a given temperature, while from 4 M, there are variations in the reduction in thickness in the course of time. For values greater than 7.5 M the flow rate decreases. 

It has been determined that it is important to obtain a constant flow that there is a sufficient active surface throughout the time, so it is preferable in this case to use sheet-like ring cans than little chips.

A theoretical model has been described to calculate the corrosion rate of the studied aluminum, which explains the reduction of the thickness of the sheet when reacting with a determined soda solution molarity in function of the temperature of the reaction, and the validation by means of the experimental tests.

We have described the effect of isopropyl alcohol which is the control of the reaction temperature below 70 °C and the filtration of the hydrogen generated, which simplifies the generation system. Purity tests and behavior analyses have been performed in other previous works in the laboratory with different types of aluminum chips, and the system generates a constant flow rate, with a purity of at least 98% with no contaminants that could damage the fuel cell, with low presence of water (>1.28%), also air if the system is not well purged and alcohol of <0.007%.

The system has been tested in a mobile machine, RC car with a 12 W fuel cell, with a low-cost generator and filter using 15 soda rings, 4 M NaOH solution and isopropyl alcohol. The performance waste aluminum to the fuel cell has been reduced by 9% compared to pure hydrogen, if we take advantage of all of the hydrogen generated. Fuel cells can supply 1.4 A at maximum power. The duration of the rings is approximately 30 min in operation, being 16 min at maximum power, but after that, the system can recharge quickly and keep working.

The advantage of this system is that you can use a low-power fuel cell machine with little cost and be able to recycle all the waste.

## Figures and Tables

**Figure 1 materials-14-07323-f001:**
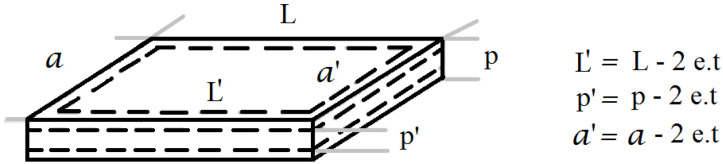
Theoretical representation of corrosion rate of a sheet by the reaction of aluminum–water.

**Figure 2 materials-14-07323-f002:**
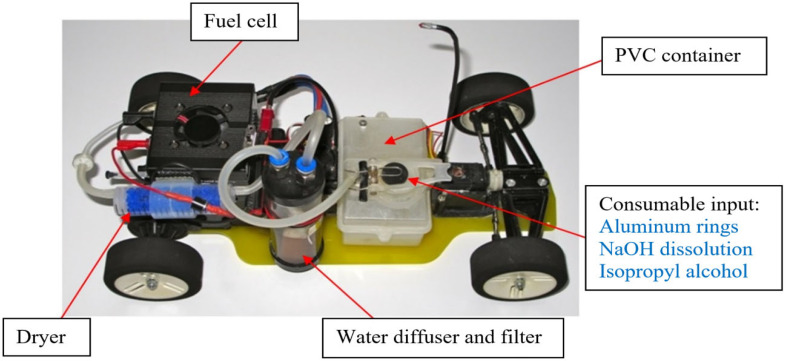
Hydrogen generation, filtering, and drying areas for the fuel cell of an RC car. (Photography: Aleix Llobet).

**Figure 3 materials-14-07323-f003:**
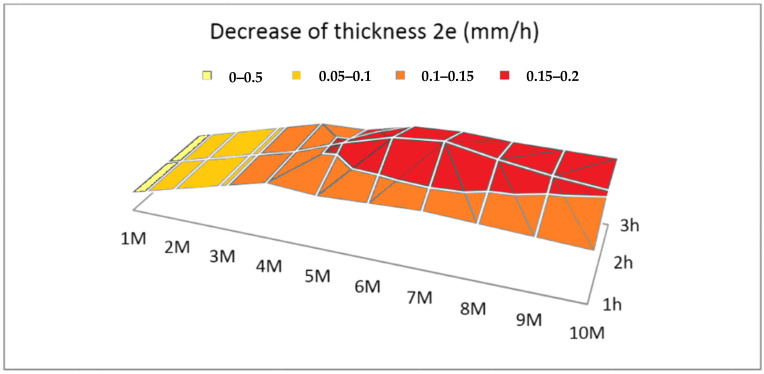
Decrease in the thickness (2*e*) of an aluminum foil (mm/h) at different molarities, at 25 °C, for 3 h.

**Figure 4 materials-14-07323-f004:**
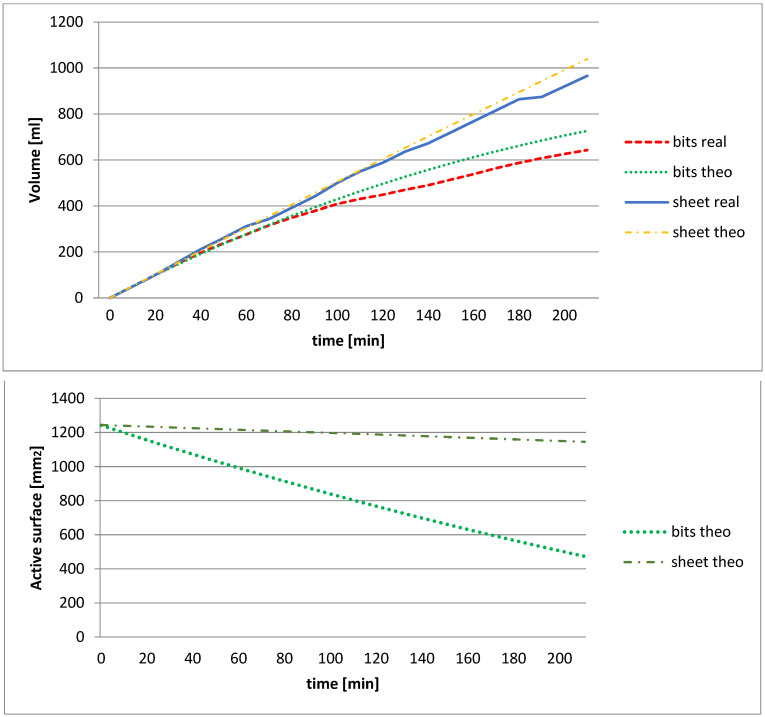
Comparison of the volume of hydrogen generated (in the figure **above**) from 104 sheet metal chips (chips) and a sheet (sheet), with the similar beginning active surface NaOH solution in a 7.5 M (with isopropyl alcohol), at 25 °C and its theoretical active surface as a function of time (in the figure **below**).

**Figure 5 materials-14-07323-f005:**
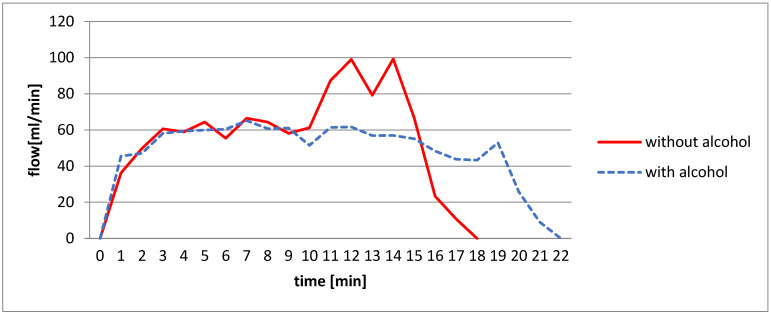
Hydrogen flow generated comparison *Q*_(*H*_2_,*t*)_ from a 7.5 M sheet, with and without isopropyl alcohol at initially 60 °C.

**Figure 6 materials-14-07323-f006:**
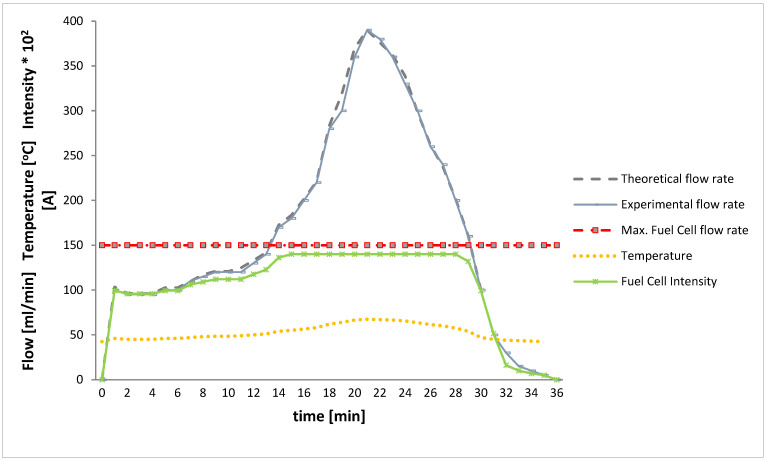
Theoretical hydrogen flow comparison of *Q*_(*H*_2_,*t*)_ produced from 15 rings and those obtained in the vehicle tank under the test conditions. The intensity generated by the fuel cell during the test is also shown.

**Table 1 materials-14-07323-t001:** Thickness reduction per surface *e* at 7.5 M as a function of temperature.

Temperature (°C)	Theoretical Reduction of Average Thickness (mm/min)	Experimental Reduction of Average Thickness (mm/min)
25	0.00125	0.00117 ± 0.00003
40	0.00370	0.00365 ± 0.00003
60	0.01410	0.01473 ± 0.0003

**Table 2 materials-14-07323-t002:** Thickness reduction per surface *e* at 4 M as a function of temperature.

Temperature (°C)	Theoretical Thickness Reduction (mm/min)	Experimental Thickness Reduction (mm/min)
50	0.00515	0.00516 ± 0.00003
60	0.00986	0.00987 ± 0.00003

## Data Availability

Not applicable.

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
