# Peer review of "Waste Aluminum Application as Energy Valorization for Hydrogen Fuel Cells for Mobile Low Power Machines Applications"

_materials, 2021, doi:10.3390/ma14237323_

Round 1

Reviewer 1 Report

  1. The authors should polish their manuscript by proficient English user
  2. The abstract and conclusions parts are not well written
  3. The authors should improve their literature review by adding more recent references 

Author Response

Answer for reviewer 1

Thanks for your comments and suggestions.

  1. The authors should their manuscript by proficient English user.

We are agreeing with the reviewer, so we have revised the text and rewritten some parts of it.

  1. The abstract and the conclusions are not well written.

We have rewritten the abstract and the conclusions and now the conclusions are supported by the results obtained in the whole research experience.

  1. The authors should improve their literature review by adding more recent references.

We have rewritten and extended the introduction and updating the references, also we have replaced some references with more recent ones in the rest of the document.

Reviewer 2 Report

Ms. Ref. No.: materials-1420868

The manuscript entitled “Waste aluminum application as energy valorization for hydrogen fuel cells for a microachine application” was studied.

The paper describes developing procedure for hydrogen chemical generation with constant flow rate and high purity applicable for fuel cell usage. The manuscript needs major revision according to following comments:

  • Here the one factor at a time (OFAT) strategy was applied for optimizing the variables. Why was “a design of experiments (DOE) strategy” not used? Notably, in OFAT strategy, the interactions among factors are not evaluated.
  • In case of Fig. 5, why were experiments performed without alcohol and at 25 0c? Why were optimized conditions not applied?
  • In case of RC car tests, how the temperature remained constant during fuel cell working?
  • The output of present work should be compared with previously reported projects and advantages and disadvantages of present work compared to others should be described in a paragraph at the end of results section

  • References should be updated. Papers published recently in 2021, 2020, 2019 should be evaluated and appropriate papers should be cited.

  • English needs revision. There are some badly constructed phrases in abstract and introduction parts. The abstract should be revised. The sentences need grammatical correctness. For examples: “That is why aluminum sheets from beverage cans are used to generate hydrogen, which will then be used as energy for the application. At the same time, a model for the flow of hydrogen has been developed at a theoretical level, considering two key variables as the temperature of the alkaline solution used and the molarity of the solution”, this part should be rewritten with appropriate description.

 or

In abstract, line 9, “fuel from the vehicle's fuel cell” should be corrected as “fuel for the vehicle's fuel cell”.

 Introduction, Second page, second paragraph, the description is not clear and should be rewritten. English needs revision.

  • In section 3.1.4, page 8, line 294, correct grammatically the sentence “it does not the flow rate increases but remains constant”

Author Response

Answer for reviewer 2

Thanks for your comments and suggestions.

  • Here the one factor at a time (OFAT) strategy was applied for optimizing the variables. Why was “a design of experiments (DOE) strategy” not used? Notably, in OFAT strategy, the interactions among factors are not evaluated.

Certainly, if the objective of the study were to study know the thickness depletion as a function of time, temperature, and molarity and the parameters following the same behavior in the study area, DOE could be applied. But, as can be seen in figure 4 the model of decrease of thickness in the function of time changes according to the molarity. As our objective is to maximize the hydrogen flow rate, and the temperature always behaves in the Arrhenius equation, we prefer to find optimal molarity and the model to said molarity. We have explained it in section 2.3 in the manuscript

(DOE is a good strategy, we have made DOE in machining tests and safe a lot of time)

  • In the case of Fig. 5, why were experiments performed without alcohol and at 25 0C?

Why were optimized conditions not applied?

Sorry. It’s an error when translating. The experiments were with alcohol. We have modified it in the manuscript. In any case, as can be seen in figure 6, if the temperature is the same, how the alcohol remains floating above the solution (less density), does not influence.

  • In the case of RC car tests, how the temperature remains constant during fuel cell

working?

             The temperature remained constant in the reactor because it works the same way

as an earthenware pitcher but with alcohol. When the alcohol evaporates it cools the liquid.

             But this system works without a condenser and there is a moment when there is little

 alcohol left and that is why temperature rises a lot. It doesn't understand

anything in the text. We modified it.

  • The output of present work should be compared with previously reported projects and advantages and disadvantages of present work compared to others should be described in a paragraph at the end of the results section

We have changed this reference to another place, with more meaning. We included the novelty of this article in the conclusions. A novelty is the application in a real vehicle, with ring cans, without a condenser, low rate flow, and will prove that the model is valid and we feed a fuel cell. In the other article, we worked in a laboratory reactor with aluminum chips, with a spacious place with a condenser but without a fuel cell. The first fuel cell we used without alcohol is out of order.

  • Reference should be updated

You’re right. We have rewritten and extended the introduction and updated the references. We have replaced some references with more recent ones in the rest of the document.

  • English needs revision. There are some badly constructed phrases in abstract and introduction parts. The abstract should be revised. The sentences need grammatical correctness. For example: "That is why aluminum sheets from beverage cans are used to generate hydrogen, which will then be used as energy for the application. At the same time, a model for the flow of hydrogen has been developed at a theoretical level, considering two key variables as the temperature of the alkaline solution used and the molarity of the solution", this part should be rewritten with appropriate description.

Thanks. Of course, we have modified it and improved the English in the whole manuscript.

  • In abstract, line 9, “fuel from the vehicle's fuel cell” should be corrected as “fuel

for the vehicle's fuel cell”. Introduction, second page, second paragraph, the description is not clear and should be rewritten. English needs revision.

We have modified the whole abstract and introduction completely.

  • In section 3.1.4, page 8, line 294, correct grammatically the sentence “it does not the flow rate increases but remains constant”

Ok, we have changed it.

Reviewer 3 Report

  1. It would be better if authors rephrase the title, title should be crisp.
  2. Improvement in the presentation of Abstract is required, it should be specific and pointed. 
  3. The keywords should be shorter, avoid long phrases. 
  4. Introduction require improvement, please provide recent references. 
  5. How this work is different from the work presented in reference 14 and 21? 
  6. Please briefly explain the recent development in the similar area of research. 
  7. What is the novelity in the submitted manuscript, please provide specific distinction from other published work on similar research work. 
  8. In the section 2.1, 2nd paragraph, line 81 to 86 on page 2 is not clear. Please explain properly. 
  9. Please correct the chemical formulas  in the text  on page 3, line 93, 95,96 and 105. Please correct throughout the Manuscript. 
  10. The corrosion rate as mentioned in fig. 1 must be explained before using it in figure caption.
  11. The unit of frequency coefficient must be properly denoted-line 159, page 4, please check throughly the manuscript. 
  12. Sec 2.3.1- please provide the results correspond to different molarity of NaOH -page 5.
  13. Please correct the line 233 on page 6.
  14. Please correct the fig. No. In the caution page 9.
  15. The conclusion require major changed be spcific, stick to result and focus on the targeted achievement of the manuscript as presented in result and discussion. 
  16. Recent litereaure survey is required to understand the current status of the presented work  and it's relevence and future scope. 
  17. Whether long duration performance testing has been done or not, please comment  on the stability with respect of time and performance loss during accelereted testing. 

Author Response

Answer for reviewer 3

Thanks for your comments and suggestions.

  1. It would be better if authors rephrase the title, title should be crisp.

You’re right. I suppose that micromachines can confuse the reader. We have changed it for: WASTE ALUMINUM APPLICATION AS ENERGY VALORIZATION FOR HYDROGEN FUEL CELLS FOR MOBILE LOW POWER MACHINES APPLICATIONS

  1. Improvement in the presentation of Abstract is required, it should be specific and pointed. 

Of course, We have modified the abstract.

  1. The keywords should be shorter, avoid long phrases. 

Ok, we have modified the keywords to get shorter.

  1. Introduction require improvement, please provide recent references. 

We have rewritten and extended the introduction and updated the references. We have replaced some references with more recent ones in the rest of the document.

  1. How this work is different from the work presented in reference 14 and 21? 

We have removed the patent references at the request of another reviewer. The difference between this work concerning the previous ones is:

  1. In the previous ones, we assay with chips from machining (aeronautics and car industries) and in this one with ring soda cans. The activation energy is different.
  2. In the previous ones, the molarity study was not shown.
  3. In the previous ones we work in a laboratory (stationary system) and with a condenser and in this case (mobile system) with an RC car system without a condenser.
  4. In the previous ones, we did not work with a hydrogen fuel cell, and in the present experience, we work with hydrogen fuel cells.

In both cases, we have validated the theoretical equations although the shape and the size are different.

  1. Please briefly explain the recent development in a similar area of research.

Recent developments work with other types of materials of hydrogen production, like ammonia borane, and other materials, and also with aluminum, but the novelty is the application of isopropyl alcohol in the production system of aluminum in the process of energy valorization.

  1. What is the novelty in the submitted manuscript, please provide specific distinction from other published work on similar research work. 

Other research articles in which on the side a hydrogen generator system they fed a 30W fuel cell (with ammonia borane), but in laboratory kit due to the complexity, we have added it in the introduction, but never an aluminum-water system. One of the problems of aluminum-water generators, when we want to introduce a machine with a fuel cell, is hydrogen filtration and temperature control, It’s difficult to place a refrigerator system in a small vehicle that maintains a constant temperature. The novelty is the alcohol use, the demonstration application, and the model to calculate the aluminum needed.

  1. In the section 2.1, 2nd paragraph, line 81 to 86 on page 2 is not clear. Please explain properly. 

We have explained it better. We have modified the second part because the authors of the referred article have published recently a correction of this data. Thanks, we did not know.

  1. Please correct the chemical formulas in the text on pages 3, lines 93, 95,96 and 105. Please correct throughout the Manuscript.

Ok, this formula has problems due to converting to pdf, we have corrected it. We have corrected the English grammar.

  1. The corrosion rate as mentioned in fig. 1 must be explained before using it in figure caption.

We have corrected it and we have explained before fig 1. In section 2.2.2

  1. The unit of frequency coefficient must be properly denoted-line 159, page 4, please check thoroughly the manuscript. 

Ok, we have corrected it.

  1. Sec 2.3.1- please provide the results corresponding to different molarity of NaOH -page 5.

Ok, We have modified section 3.1.1 so that it is understood. Like We have worked with 4M we have added corrosion rate at section 3.1.3.

  1. Please correct the line 233 on page 6.

Ok, thank you for the comment

  1. Please correct the fig. No. In the caution page 9.

Ok, thanks.

  1. The conclusion requires major changed be specific, stick to result and focus on the targeted achievement of the manuscript as presented in result and discussion. 

Ok, we have modified conclusions as presented in the result and discussion.

  1. Recent literature survey is required to understand the current status of the presented work and it's relevance and future scope. 

We have modified and improved the introduction and updated the references.

  1. Whether long duration performance testing has been done or not, please comment on the stability with respect of time and performance loss during accelerated testing. 

We have commented on the results in the form of the performance of fuel cells, always in the zone where the hydrogen flow rate generate is the least that required by the fuel cell. The hydrogen obtained with the aluminum-water system reaction has a performance about of 98%. Balance is performance aluminum-fuel cell power 93%, total balance 91%. In this RC car, the fuel cell outlet must be left open so that excess hydrogen is lost and it has a lower performance but could be stored in the circuit by pressure.

Round 2

Reviewer 2 Report

The revised manuscript is well. However, the new paragraph added to the end of section 2.1 needs English language editing. it should be corrected grammatically.

Author Response

REVIEWER 2

The revised manuscript is well. However, the new paragraph added to the end of section 2.1 needs English language editing. it should be corrected grammatically.

We have performed a complete revision of English in the manuscript uploaded. The changes are red in the provided version

Reviewer 3 Report

Please check the sentences and appropriately rephrase throughout the Manuscript.

Please make the uniform formatting throughout the Manuscript.

It would be better if Authors specifically mention the novelty of present work in the last paragraph of introduction.

Author Response

REVIEWER 3

Please check the sentences and appropriately rephrase throughout the Manuscript.

We have performed a complete revision of English in the manuscript uploaded. The changes are red in the provided version

Please make the uniform formatting throughout the Manuscript.

We have performed a revision and uniform of formats throughout the manuscript

It would be better if Authors specifically mention the novelty of present work in the last paragraph of introduction.

Thanks for comment. We have added a paragraph with the novelty of the present work, highlighted in red in the provided version